# Location of Recurrences after Trimodality Treatment for Glioblastoma with Respect to the Delivered Radiation Dose Distribution and Its Influence on Prognosis

**DOI:** 10.3390/cancers15112982

**Published:** 2023-05-30

**Authors:** Nika Guberina, Florian Padeberg, Christoph Pöttgen, Maja Guberina, Lazaros Lazaridis, Ramazan Jabbarli, Cornelius Deuschl, Ken Herrmann, Tobias Blau, Karsten H. Wrede, Kathy Keyvani, Björn Scheffler, Jörg Hense, Julian P. Layer, Martin Glas, Ulrich Sure, Martin Stuschke

**Affiliations:** 1Department of Radiation Therapy, West German Cancer Center, University of Duisburg-Essen, University Hospital Essen, 45147 Essen, Germany; 2Department of Neurology, University of Duisburg-Essen, University Hospital Essen, 45147 Essen, Germany; 3Department of Neurosurgery and Spine Surgery, West German Cancer Center, University of Duisburg-Essen, University Hospital Essen, 45147 Essen, Germany; 4Institute of Diagnostic and Interventional Radiology and Neuroradiology, University of Duisburg-Essen, University Hospital Essen, 45147 Essen, Germany; 5Department of Nuclear Medicine, University Duisburg-Essen, University Hospital Essen, 45147 Essen, Germany; 6Institute of Neuropathology, University of Duisburg-Essen, University Hospital Essen, 45147 Essen, Germany; 7German Cancer Consortium (DKTK), Partner Site University Hospital Essen, 45147 Essen, Germany; 8DKFZ-Division Translational Neurooncology at the West German Cancer Center (WTZ), DKTK Partner Site, University Duisburg-Essen, University Hospital Essen, 45147 Essen, Germany; 9Department of Medical Oncology, West German Cancer Center, University of Duisburg-Essen, University Hospital Essen, 45147 Essen, Germany; 10Department of Radiation Oncology, University of Bonn, University Hospital Bonn, 53127 Bonn, Germany; 11Institute of Experimental Oncology, University of Bonn, University Hospital Bonn, 53127 Bonn, Germany

**Keywords:** recurrence pattern, glioblastoma, intensity modulated radiation therapy (IMRT), volumetric modulated arc therapy (VMAT), neurosurgery

## Abstract

**Simple Summary:**

This scientific research characterizes the recurrence pattern of patients with glioblastoma after trimodality treatment, neurosurgery and radiation therapy, either intensity modulated radiation therapy (IMRT) or volumetric intensity modulated arc therapy (VMAT), as well as concurrent chemotherapy in a clinical setting based on daily image-guided radiation therapy (IGRT) at the West German Cancer Center. This study underlines the importance of reconsideration of clinical target volume margins in the clinical routine. Larger radiation therapy margins may decrease the proportion of out-field recurrences, but an effect on overall survival is highly questionable.

**Abstract:**

Background: While prognosis of glioblastoma after trimodality treatment is well examined, recurrence pattern with respect to the delivered dose distribution is less well described. Therefore, here we examine the gain of additional margins around the resection cavity and gross-residual-tumor. Methods: All recurrent glioblastomas initially treated with radiochemotherapy after neurosurgery were included. The percentage overlap of the recurrence with the gross tumor volume (GTV) expanded by varying margins (10 mm to 20 mm) and with the 95% and 90% isodose was measured. Competing-risks analysis was performed in dependence on recurrence pattern. Results: Expanding the margins from 10 mm to 15 mm, to 20 mm, to the 95%- and 90% isodose of the delivered dose distribution with a median margin of 27 mm did moderately increase the proportion of relative in-field recurrence volume from 64% to 68%, 70%, 88% and 88% (*p* < 0.0001). Overall survival of patients with in-and out-field recurrence was similar (*p* = 0.7053). The only prognostic factor significantly associated with out-field recurrence was multifocality of recurrence (*p* = 0.0037). Cumulative incidences of in-field recurrences at 24 months were 60%, 22% and 11% for recurrences located within a 10 mm margin, outside a 10 mm margin but within the 95% isodose, or outside the 95% isodose (*p* < 0.0001). Survival from recurrence was improved after complete resection (*p* = 0.0069). Integrating these data into a concurrent-risk model shows that extending margins beyond 10 mm has only small effects on survival hardly detectable by clinical trials. Conclusions: Two-thirds of recurrences were observed within a 10 mm margin around the GTV. Smaller margins reduce normal brain radiation exposure allowing for more extensive salvage radiation therapy options in case of recurrence. Prospective trials using margins smaller than 20 mm around the GTV are warranted.

## 1. Introduction

The optimal target volume for radiotherapy of glioblastoma is unknown. Moreover, European and US target delineation concepts differ significantly [1,2]. According to European delineation concepts, a one-phase radiotherapy series is used. For the clinical target volume (CTV) a standard margin of 20 mm around the residual gross tumor, defined as the contrast-enhancing mass on T1-weighted MRI or the resection cavity after surgery, is applied with respect to anatomic borders. On the contrary, North American cooperative groups often treat patients in two phases instead of one with shrinking target volumes [3,4]. The gross tumor here is defined as the tumor-associated abnormalities in post-contrast T1 as well as the tumor-surrounding oedema found on FLAIR/T2 sequences. In the first phase up to a cumulative dose of 46 Gy, both the T1 post-contrast and T2 hyperintensities are included into the target plus a margin of 20 mm and if no oedema exists, a CTV margin of 2.5 cm is used. These margins are reduced around anatomic barriers to tumor growth such as falx, skull or ventricles. Accordingly, the target volume in the first phase can be much larger than that following the European consensus. In the second phase, the target volume is similar to the target volume proposed by the European consensus [1]. The inclusion of the oedema is based on stereotactic serial biopsy findings that showed isolated tumor cells in the area of hyperintensities on T2-weighted MRI [5]. Moreover, validated prognostic factors for individualization of the target volumes are not existent. As a consequence of the standard margins, statistically more than 50% of the ipsilateral hippocampus overlaps with the planning target volume (PTV) [6], risking neurocognitive impairment [7]. The prognosis of patients with glioblastoma improved with the introduction of radiotherapy. However, despite large CTV margins and new emerging treatment options, such as tumor-treating fields, local relapses are ubiquitously present [8]. In addition, studies using smaller CTV margins found a similar rate of central or in-field recurrences compared to studies with larger margins using two-phase target volume delineation [4,9,10,11,12,13]. Nevertheless, the use of conventional margins persisted in the clinical routine. In this study, we performed a competing-risks analysis of local recurrences in dependence on their location around the gross tumor volume after standard margin radiotherapy of glioblastoma patients plus temozolomide if indicated. The results obtained are used to quantitatively estimate the effectiveness of additional CTV margins beyond the 10 mm CTV margin around the gross target volume on survival.

## 2. Materials and Methods

Based on a systematic institutional database search, all consecutive patients with newly diagnosed and histopathologically confirmed glioblastoma recurrence treated at the department of radiation therapy of the West German Cancer Center with conventional or hypofractionated radiation therapy after neurosurgical resection at age > 18 years were included in the time period from 1 January 2007 to 31 December 2021. The recurrence was confirmed histopathologically, by high-resolution MRI or/and PET/MRI and follow-up of at least 3 months. If IDH-mutation status was available, IDH-mutant WHO grade 4 astrocytoma were excluded. IDH-mutation analysis was performed immunohistochemically. A supratentorial component was obligatory. Cases with history of chemotherapy prior to GBM diagnosis were excluded. The study was conducted after the local ethics committee approval (20-9719-BO).

### 2.1. Treatment Sequence, Radiation Therapy Planning and Radiation Therapy

All patients were discussed in an interdisciplinary tumor board at initial diagnosis and at time point of tumor recurrence. The post-surgery MRI was used as the baseline imaging for which response was determined as recommended by Ellingson et al. [14]. The MRI scan comprised high-resolution pre- and post-contrast 3D T1-weighted sequences (1 mm slice thickness; gadolinium-chelated contrast agent), T2/FLAIR (fluid-attenuated inversion recovery) sequences, T2-weighted turbo spin-echo (TSE)-sequences, Diffusion (b = 0, 500, 1000 s/mm^2^ ≥ 3 directions) and ADC-Maps (apparent diffusion coefficient) at a 1.5 or 3T MRI-scanner. Following the first cycles of adjuvant therapy, patients received regularly scheduled follow-up MRI scans. To differentiate tumor recurrence from post-therapeutic changes and to exclude possible confusion with pseudo-progression, we included only patients with histopathologically confirmed recurrence, or confirmed tumor progression on further follow-up MRI and/or PET/MRI. Radiation techniques changed over time. From 2000 to 2012, 3D conformal radiotherapy was used, from 2012 to 2018, static field IMRT was the preferred technique, and since 2018, non-coplanar VMAT has been the most widely applied technique. Three-dimensional planning was performed with the treatment planning system Eclipse (Varian Medical Systems). Reproducible positioning was achieved with a thermoplastic mask system. The EORTC 26981/22981 quality assurance statements were adopted [15]. Total radiation dose was 60 Gy, using conventional fractionation of 2 Gy per daily fraction, 5 fractions a week. Elderly patients >70 years or patients with a Karnofsky performance status <50 were offered a shorter hypofractionated radiation schedule with 2.67 Gy per daily fraction to 40.05 Gy [16,17]. The PTV received 95% of the prescription dose. Six MeV photons from a linear accelerator were used. Patients received concurrent chemotherapy with temozolomide in case of MGMT promoter hypermethylation or based on individual decision making depending on clinical performance and age.

### 2.2. Margin Definition and Clinical Target Volume Definition

Gross tumor volume (GTV) and clinical target volume (CTV) were defined according to the ESTRO–ACROP contouring guidelines [1]. In macroscopically resected tumors, GTV delineation was based on the resection cavity plus any residual enhancing tumor on contrast-enhanced T1 weighted MRI, without inclusion of peri-tumoral oedema. In general, GTV included all postoperative contrast-enhancing areas, detected on an early postoperative MRI performed within 72 h after brain surgery.

The CTV was defined as the GTV plus a 20 mm margin to account for microscopic spread along the white matter tracts with respect to anatomical barriers. T2/FLAIR MRI high-signal regions were included if they were considered to represent regions of low-grade tumor. In addition, in some patients peritumoral oedema was included up to a margin of about 3.0 cm around the GTV. A planning target margin of 5 mm was used.

### 2.3. Identification of Recurrence and Recurrence Pattern

Recurrence patterns were assessed as in-field or out-field as previously described [18,19,20]. Isodose curves encompass a particular area of absorbed dose. Recurrences were classified as in-field, if >80% of the tumor recurrence resided within the prescription 95% isodose, marginal if 20–80% of the tumor recurrence resided within the 95% isodose, and out-field if <20% of the tumor recurrence resided inside the 95% isodose [20]. In addition, the location of the recurrence in relation to the GTV after surgery was quantified by the percentage overlap of the volume of the recurrence with the GTV, isotropically expanded by 10 mm, 15 mm and 20 mm, resulting in three CTVi-volumes, CTV10mm, CTV15mm and CTV20mm. Recurrences were classified as in-field with respect to the annotated margin around the postoperative GTV, if >80% of their volume overlapped with the postoperative GTV expanded by a margin of 10 mm, 15 mm, 20 mm, or the 95% or 90% isodose surface of the delivered dose distribution. The recurrent tumor volume was delineated in the treatment planning system after fusion of the respective MRI or/and PET/MRI which showed the recurrence within the original planning CT.

### 2.4. Statistics, End-Points

Descriptive statistics and statistical analysis were performed with SAS (version 14.3, SAS Institute, Cary, NC, USA). The Friedman’s and Cochran’s Q test were performed using the procedure FREQ, survivor function analysis was performed using the procedure LIFETEST, and proportional hazard analyses of survival times were performed using the procedure PHREG. Association of in- or out-field recurrences with patient, tumor or treatment classification variables was performed using the procedure FREQ and with continuous variables using logistic regression (PROC LOGISTIC). Friedman’s and Cochran’s Q-tests were performed with the procedure FREQ.

Cumulative incidence functions as subdistribution functions for recurrences at the given locations were calculated using the procedure LIFETEST. The cumulative incidences can be interpreted as estimates of the risk of observable cumulative occurrence of the considered type of event as the first event under concurrent risks. The Kaplan–Meier method estimates the underlying cumulative probability of an event by time t where the other types of events are removed, assuming non-informative censoring, i.e., independence between the competing risks events [21]. Here, events other than the event type under consideration were treated as censoring events. The procedure LIFETEST from SAS was used. These Kaplan–Meier probabilities of relapses according to location over time were used for the random number calculations assuming independence of the risks of recurrences at the different locations.

Random number simulations were performed to simulate the effect of margin reduction from a conventional clinically applied margin obtained from the 95% isodose line to a 10 mm margin around the CTV. The survival times using a conventional margin were obtained as the sum of two random observations from exponential distributions for random disturbance, the times to recurrences and the survival time after recurrence (function RAND from SAS). For each simulated observation, three independent failure times were simulated, the time to in-field recurrence within a 10 mm margin around the CTV, or within a margin of >10 mm up to the 95% isodose, or out-field the 95% isodose of the clinical plan. The scale parameters for the exponential distributions were obtained by fitting a parametric model to the failure time data to first recurrence using log-transformed failure times and an exponential (procedure LIFEREG from SAS). The recurrence location with the shortest associated failure time was taken as the observed recurrence for this observation. The survival times using a reduced margin of 10 mm were obtained in the same way but using a reduced median time to recurrences located beyond 10 mm and up the 95% isodose of 5.5 months or 12 months. For out-field recurrences beyond the 95% isodose, a reduction of a delay of 2.8 months or 5.6 months was assumed. Survival after recurrence was obtained by another random number failure time from an exponential distribution with a scale parameter obtained from the curves after survival recurrence. Simulations for trials comparing a CTV margin of 10 mm with conventional margins as used in this study were repeated 100 times with 500 patients and the power to detect significant differences was calculated.

Tumor recurrence was defined according to the RANO criteria [22] by means of MRI, PET/MRI and/or by histopathological confirmation. Post-recurrence survival was defined as time from recurrence to death of any cause or last follow-up imaging for living patients. Overall survival was measured from start of radiotherapy to death or last MRI.

## 3. Results

Altogether 91 consecutive patients (41 female, 50 male) with glioblastoma grade IV diagnosis fulfilled the inclusion criteria of this study (Table 1). Forty-seven patients received a gross total resection while 44 underwent a partial resection. Median time from neurosurgery to radiation therapy start was 28.0 days. Eighty-five patients received conventional fractionation and six hypofractionation at 5 × 2.67 Gy ad 40.05 Gy. Radiation therapy series lasted in median 42 days (Interquartile range (IQR) 41–43 days) for patients receiving conventional and 20 days (IQR 20–21 days) for those receiving hypofractionation. Seventy-seven patients received temozolomide (75 mg/m^2^ daily) simultaneously to radiation therapy, while five patients received doublet chemotherapy with temozolomide/lomustine. MGMT promoter hypermethylation status was present in 30 patients. Additional patient, tumor and treatment characteristics are summarized in Table 1. IDH-mutation status could not be obtained for 20 tumors, as histopathologic probes were no longer available. Survival of patients with IDH wild type glioblastomas and the patients with unknown IDH-mutation status had the same survival, confirming the homogeneity of prognosis according to this parameter (Appendix A). Median follow-up with MRI-imaging from start of radiotherapy was 14.9 months (IQR 8.5–23.3 months). Median times from start of radiotherapy to recurrence on imaging was 5.3 months (IQR 3.0–11.3 months), which was confirmed either histopathologically or based on follow-up MRI or PET/MRI. The recurrence pattern was either multifocal or unifocal, i.e., located in one brain region or affecting multiple regions. The validity of the diagnosis of recurrence was further assessed by analysing patients with very small recurrence volumes < 1 cm^3^ at the time of detection of recurrence. Of those patients with very small recurrence volumes < 1 cm^3^ (25 patients), all received additional high-resolution MR-follow-up imaging with confirmation of tumor progress, 36% received a PET/MRI confirming tumor recurrence and 52% received a histological confirmation (salvage re-resection). Median survival of all patients was 18.7 months (95% CI: 15.3–24.2 months).

The median distance between the GTV and the 90% isodose was 27.0 mm (IQR: 24.0–30.0 mm), as obtained by the differences of the radii of the volume equivalent spheres of the 90% isodose and the GTV. The respective median margin between the GTV and the 95% isodose was 25.5 mm (interquartile range: 22.5–28.0 mm). Altogether 58, 62, 64, 80, and 80 of the 91 patients fulfilled the criteria of in-field recurrences within a margin of 10 mm, 15 mm, 20 mm, or the 95% or the 90% isodose around the postoperative GTV (Figure 1). In addition, 18, 17, 17, 5, and 5 patients had marginal recurrences within a margin of 10 mm, 15 mm, 20 mm, the 95% or the 90% isodose around the postoperative GTV (Figure 2). Out-field recurrences with respect to the 95% isodose occurred in 11 patients. The frequencies of in-field recurrences were significantly dependent on the margin around the GTV (Cochran’s Q-test for this repeated measures design, *p* < 0.0001). Comparing the proportions of in-field recurrences at the three margins of 10 mm, 15 mm and 20 mm around the GTV alone, the dependence of the probability of in-field recurrences on the margin remained significant (*p* = 0.0094, Cochran’s Q-test). In more detail, Figure 2 shows the cumulative proportions of the recurrences overlapping with the target volume by the partial volumes smaller or equal to the values indicated on the *x*-axis. Target volumes were expanded by margins of 10 mm, 15 mm and 20 mm around the GTV or up to the 95% or 90% isodose of the delivered dose distribution, respectively. In addition, the partial volume of the recurrence overlapping with the target volume stratified by patient differed significantly between the margins around the postoperative GTV (*p* < 0.0001, Friedman’s-Test).

This study analyzed patients with documented progression on MRI during follow-up. However, almost all glioblastoma progress during follow-up [16,17], and therefore, overall survival can only be improved by a target volume expansion, if overall survival for patients with in-field recurrences is markedly longer than that for patients with out-field recurrences or if the out-field recurrence probability is high. We tested this hypothesis and found no influence of location of recurrence on overall survival (Figure 3).

Median survival was 18.5 months (95% CI: 15.0–25.3 months) in the group of 58 patients with tumors recurring inside a margin of 10 mm around the GTV. It was 20.1 months (95% CI; 12.5–26.8 months) for 24 patients with tumors fulfilling the in-field recurrence criterion inside the 95% isodose of the delivered dose distribution, but outside a margin of 10 mm, and it was 18.7 months (95% CI: 13.0–24.2 months) for 11 patients with tumors recurring outside the 95% isodose (*p* = 0.7053, log rank test). Figure 4a shows the Kaplan–Meier curves for freedom from recurrence within a given margin location of recurrence. Curves differ significantly according to location of recurrence fulfilling the in-field criterion at margins of 10 mm at a minimum margin > 10 mm but within the 95% isodose or outside the 95% isodose (*p* < 0.0001, Cochrane’s Q-test comparing the frequencies of relapses according to location). The Kaplan–Meier probabilities of relapsing at the different locations within 24 months are 78.2% (95% CI: 66.4–88.9%) for margins of <10 mm, 53.6% (95% CI: 33.2–76.8%), for a minimum margin > 10 mm, but within the 95% isodose, or 34.8% (95% CI: 18.5–58.9%) outside the 95% isodose.

Figure 4b highlights observed cumulative incidences of recurrences according to the margin of recurrences considering recurrences at the different locations as concurrent risks (*p* < 0.0001). Cumulative incidences of in-field recurrences at 24 months were 60%, 22% and 11% for recurrences located within a 10 mm margin, outside a 10 mm margin but within the 95% isodose, or outside the 95% isodose using competing risk estimates (*p* < 0.0001, Cochrane’s Q-test comparing the frequencies of relapses according to location alone).

Table 2 summarises patient, clinical and treatment-related characteristics in relation to the occurrence of an out-field recurrence. The only prognostic factor significantly associated with out-field recurrence was multifocality of recurrence (*p* = 0.0037 Fisher’s Exact Test). In relation to the minimum margin around the GTV required to fulfil the in-field criterion, four of 58 (7%) recurrences at 10 mm were multifocal, while six of 22 (27%) recurrences at minimum margins >10 mm but < the 95% isodose were multifocal. From the 14 recurrences outside the 95% isodose, seven (50%) were multifocal. These proportions differed significantly (*p* < 0.0001, Fisher’s exact test).

Next, we analysed prognostic factors related to the survival times from the diagnoses of recurrence. Time from recurrence to death was slightly shorter for out-field recurrences at the 95% isodose than for patients with in-field recurrence at a margin of 10 mm or at a minimum margin of >10 mm but within the 95% isodose of the delivered dose (*p* = 0.1714, log-rank test, Figure 5a). Multifocality of the recurrence was associated with shorter survival times after diagnosis of recurrences than unifocality (Figure 5b, *p* = 0.0294, log-rank test).

Patients with tumors amenable to reresection receiving gross total reresection had longer survival times compared to patients with partial or no resection (Figure 5c, *p* = 0.0069, log-rank test).

In a next step, we performed Monte Carlo simulations to get insights into what decrease in survival could be expected by decreasing the clinically used CTV margins related to the 95% isodoses to 1 cm margins using the Kaplan–Meier estimates of time to recurrence according to location of recurrence given in Figure 4a and the survival times after recurrences given in Figure 5a. Compared with the small-volume radiotherapy with 10 mm CTV margins, conventional-margin radiotherapy may delay recurrences meeting the in-field criterion by a median of 5.5 months for margins >10 mm and smaller than the 95% isodose, and marginal recurrences outside the 95% isodose by 2.8 months. Figure 6 depicts a representative survival curve for groups of 500 simulated individuals per margin size (*p* = 0.5333, log-rank test). The power for such a simulated trial with 2 × 500 patients was only 6% to detect the effect of increased margins at *p* = 0.05 using the log-rank test. Even if the delay of recurrence by radiotherapy is estimated to be 12 months for margins > 10 mm and smaller than the 95% isodose and 5.5 months outside the 95% isodose for conventional marginal radiotherapy compared with small-volume radiotherapy with 10 mm CTV margins, such an effect can only be demonstrated by studies of 2 × 500 patients with a power of 14% under the assumptions used. This demonstrates that under the dominant in-field recurrence risk at margins < 10 mm, as observed in this study and concurrent risks, a moderate delay of recurrences at larger margins by radiotherapy will be small and hardly detectable even by large clinical trials.

## 4. Discussion

This study shows that the concurrent risk of local recurrence within 1 cm margin to the gross tumor volume is on average dominant and occurs faster than recurrences at larger distances to the resection cavity or residual macroscopic tumor. As glioblastomas are heterogeneous, the hypothesis can be sustained from the present data that the most resistant population resides within the vicinity of the gross tumor. These results confirm the findings from retrospective trials that most recurrences are in field after postoperative first-line radiotherapy of glioblastomas using standard margins [1]. We found percentages of 62% and 85% in-field recurrences with a CTV margin of 10 mm or the 95% isodose of the clinical plan. As long as this risk remains as high as it is, therapies directed to cells more distant from the gross tumor will not be very successful.

Regarding the clinical plan, similar percentages were found by other groups [23,24,25,26,27]. With respect to the 10 mm margin, the rate of in-field recurrences in this study was similar to those in the study by Wallner et al. [28], but slightly lower than in the studies by Gebhardt et al., 2014 [9] and Buglione et al., 2016 [23] with about 80%. The cumulative incidence of in-field recurrences at larger safety margins decreases significantly, and yet out-field recurrences at larger safety margins are associated with multifocality. In addition, this study found that the prognosis of out-field recurrences tended to be worse than that of in-field recurrences from the time of diagnosis of the recurrence, which is related to the multifocality of these recurrences. Jiang et al., 2020 also found shorter post-progression survival for patients with distant or out-field progressions than for in-field progressions [29]. However overall survival from initial diagnosis was similar for in-field and out-field recurrences in the present study. Therefore, we could not validate the findings of Brandes et al. [20] of an improved survival of patients with out-field recurrences.

The detection of the extent of glioblastoma spread by imaging studies, especially the non-contrast enhancing extent, is a matter of ongoing research [30]. Clearly demonstrated, glioblastomas tend to extend beyond contrast-enhancing volumes found on T1-weighted MRI [31]. Tumor infiltration beyond gadolinium-enhancing areas can be detected by [18F]FET/PET [32]. Moreover, there exist also detection limits with 5-ALA-induced fluorescence intraoperatively. Gliolan (5-ALA) is approved in adults for the visualization of malignant tissue during surgery for malignant gliomas WHO grades III and IV. 5-ALA is a prodrug that is metabolized intracellularly to the fluorescent molecule PPIX. Tumor fluorescence is typically higher than that of normal tissue. Ideally, the high contrast allows visualization of tumor tissue under blue-violet light. However, about half of tissue samples from the tumor boundary with no fluorescence signal contained infiltrative tumor tissue [33].

Different tumor cell foci can be genetically heterogeneous, can share only half of their mutations and can acquire additional mutations by parallel genetic evolution [34]. Kiesel et al. analyzed in their prospective study tissue samples from 5-ALA fluorescence-guided glioblastoma resection with differing 5-ALA staining. Tumor cells in samples with no fluorescence showed less mitotic activity and less cell density than in 5-ALA positive zones [33]. In addition, there is evidence that 5-ALA labelling corresponds with 11C Methionine PET uptake [35]. High volume of extravascular extracellular space by T1 dynamic contrast-enhanced MRI corresponds to increased mitotic activity of glioblastoma cells [36]. In conclusion, these data support the hypothesis that the most resistant parts of the tumor are clinically detectable and are included into the target volume within 1 cm margin.

Cerebral blood volume from perfusion-weighted MRI and [18F]FET/PET-uptake correlated with neovascularization and cellularity of glioblastoma [37]. However, correlation of quantitative MRI parameters obtained 1–3 days prior to biopsy with histopathology by frame-based stereotactic biopsies found only a weak correlation between T1 relaxation times and no correlations with T1, T2, T2* and T2′ relaxation times from quantitative MRI with cell density [38]. Supratotal resection of glioblastoma beyond the contrast-enhanced T1 region might be related to improved survival, but randomised trials are lacking [39]. In one large study, the postoperative FLAIR volume was neither associated with recurrence nor with survival after gross total or supratotal resection [40].

In line with others, this study demonstrates that central in-field recurrences are the dominant cause of failure after standard postoperative radiochemotherapy of glioblastomas. The relapse pattern of glioblastomas can be moderately changed by dose escalation of radiotherapy, by temozolomide for MGMT promoter hypermethylated glioblastomas, or addition of TT-Fields towards more marginal or out-field recurrences [20,41,42]. However, a survival benefit of radiation dose escalation over conventional fractionated standard radiotherapy has not shown a survival benefit up to now in randomised trials [43,44]. Under circumstances of increased in-field control, the impact of larger margins to reduce out-field recurrences will increase.

Assuming competing risks of recurrences at the different distances from the resection cavities observed in this study and using the time-dependent probabilities for the different events reveals that an effect on survival by expanding the radiotherapy target volume beyond 10 mm will unlikely be detectable by large randomised trials. Here, a moderate recurrence-delaying effect of radiotherapy in the expanded volumes of median 5 months is anticipated. Even assuming a 12-month delay, a long-term survival benefit was barely demonstrable from this study because of the high risk of in-field recurrence with a 10 mm margin.

After neurosurgery, radiochemotherapy and chemotherapy tumor progression or recurrence may be difficult to differentiate from pseudo-progression. Therefore, a major strength of the present study was that only patients with histopathologically confirmed tumor recurrence or tumor progression confirmed on follow-up imaging, either by MRI and/or PET/MRI, were included.

In the following, a range of limitations of the present study results shall be discussed. The Kaplan–Meier probabilities of freedom from relapse at the considered localizations reflect the average for the group of patients studied. However, there may be larger inter-individual differences. Patients suffering from out-field recurrences might have a lower underlying risk of in-field recurrences with an increased gain of a larger CTV margin. This study does not allow for subgroup analysis and subgroups behave differently. However, no subgroups were excluded from this cohort, and therefore this group of patients reflects our experience in real life with patients who have close imaging follow-up studies.

## 5. Conclusions

The present results indicate that the classic GTV-CTV delineation strategy for postoperative RT of GBM should be reconsidered in favor of smaller CTV margins. While not significantly hampering tumor control, smaller CTV margins may allow for better protection of healthy tissue, thus ensuring patients’ quality of life and leaving therapeutic options for salvage RT in the yet unpreventable event of GBM recurrence. Prospective randomised clinical trials using smaller CTV margins are warranted to confirm these findings.

## Figures and Tables

**Figure 1 cancers-15-02982-f001:**
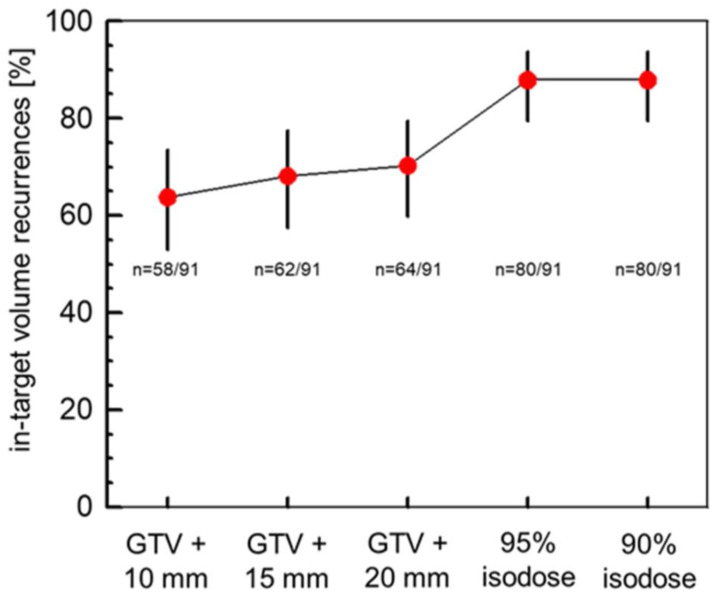
Proportions of in-target volume recurrences in dependence on the margins around the postoperative gross target volume. GTV: postoperative gross target volume; binominal proportions of in-field recurrences are given with the exact Clopper–Pearson 95% confidence intervals; n: proportion of in-field recurrences enclosed by the indicated margin around the postoperative GTV. The hypothesis that the probability of in-field recurrences did not depend on the margin around the GTV could be rejected using Cochran’s Q-test for this repeated measures model (*p* < 0.0001).

**Figure 2 cancers-15-02982-f002:**
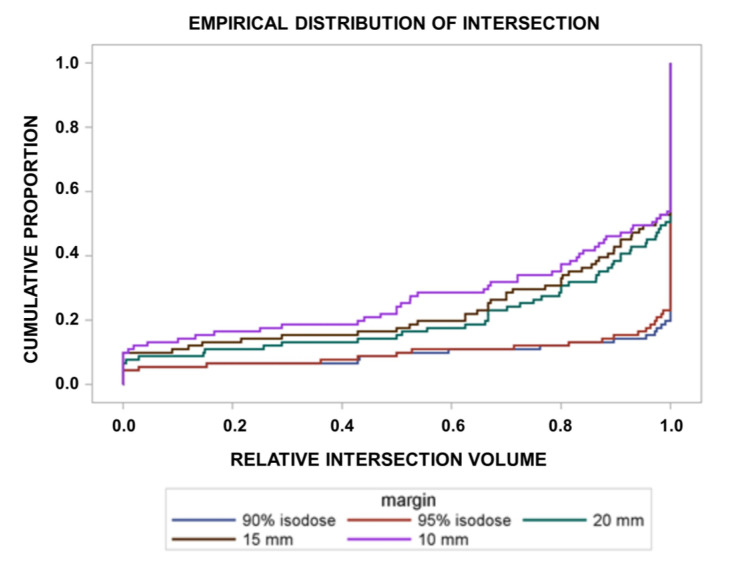
Empirical distribution plot of the fraction of the recurrence within the target volume over the 91 patients in dependence of the margin around the postoperative GTV. Cumulative proportions of the recurrences intersecting with the target volume by partial volumes smaller or equal to the values indicated on the *x*-axis, as a function of the margins around the GTV. Margin 10 mm, 15 mm, 20 mm, 95% isodose, 90% isodose: target volumes were expanded by margins of 10 mm, 15 mm, 20 mm around the postoperative GTV, or up to the 95% or to the 90% isodose levels of the delivered dose distribution, respectively (*p* < 0.0001, Friedman’s-Test for repeated measures design).

**Figure 3 cancers-15-02982-f003:**
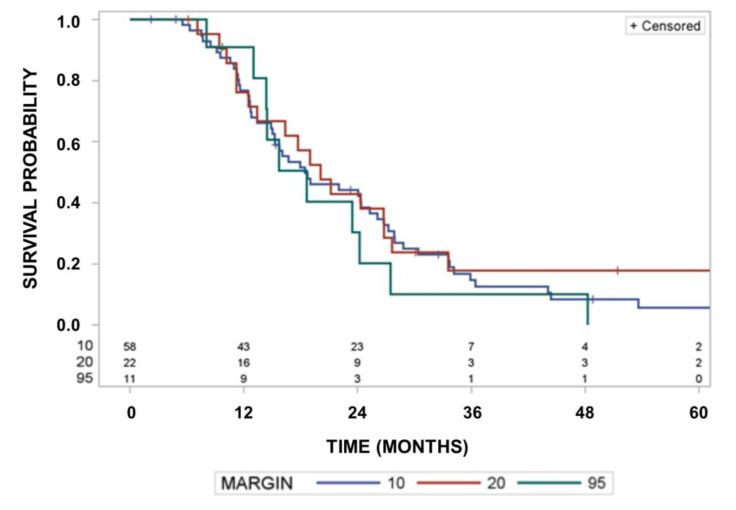
Survival curves from the start of radiotherapy in dependence on location of the recurrence. Survival curves as a function of the minimum margin around the GTV at which the in-field recurrence criterion is fulfilled. Margin = 10: in-field recurrence criterion fulfilled at a margin of 10 mm; Margin = 20: in-field recurrence criterion fulfilled at a minimum margin of >10 mm but within the 95% isodose of the delivered dose; Margin = 95: patients with recurrences outside the 95% isodose. Survival curves did not differ (*p* = 0.7053, log rank test).

**Figure 4 cancers-15-02982-f004:**
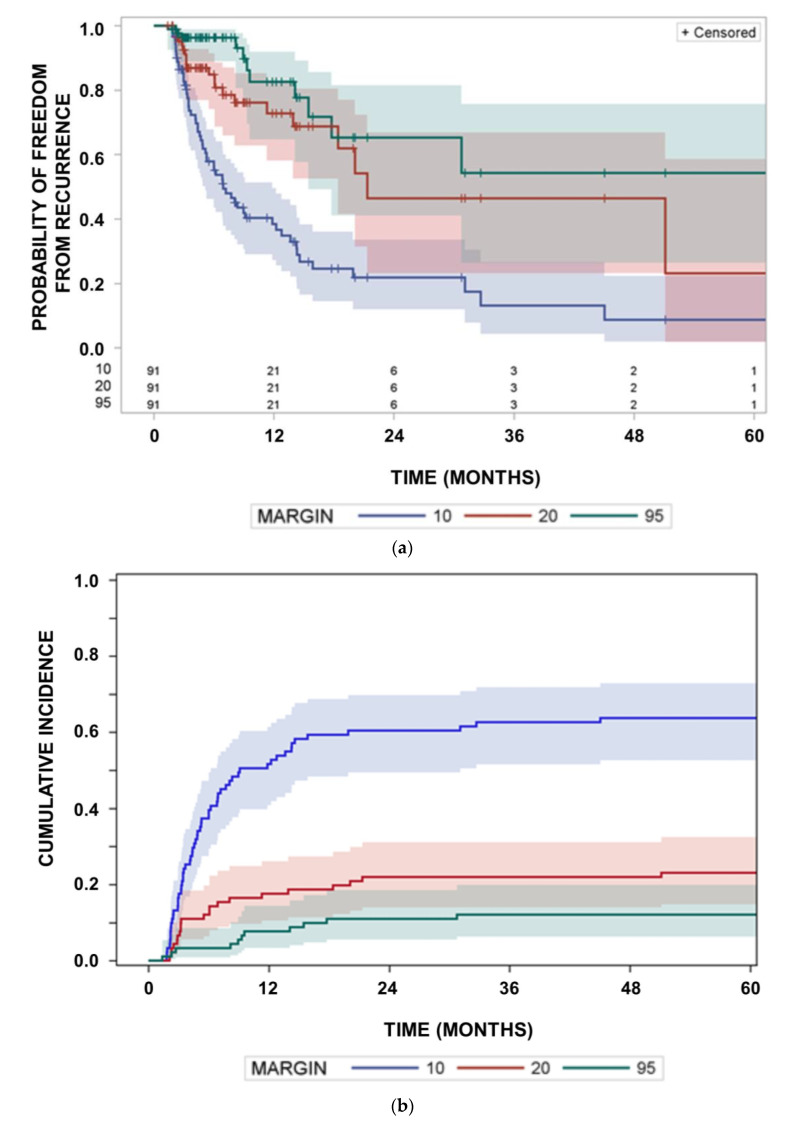
(**a**) Kaplan–Meier curves of probability of freedom of recurrence according to location of recurrence. Comparison of freedom from recurrence curves as a function of the margin of recurrences. Curves differ significantly between margin 10, 20 and 95 (*p* < 0.0001, Cochrane’s Q-test comparing the frequencies of relapses according location alone). (**b**) Cumulative incidences of recurrences according to location of recurrence. Comparison of cumulative recurrence incidences as a function of the recurrences. Margin = 10: in-field recurrence criterion fulfilled at a margin of 10 mm; Margin = 20: in-field recurrence criterion fulfilled at a minimum margin of >10 mm, but within the 95% isodose of delivered dose; Margin = 95: patients with recurrences outside the 95% isodose (*p* < 0.0001, Cochrane’s Q-test comparing the frequencies of relapses according to location alone).

**Figure 5 cancers-15-02982-f005:**
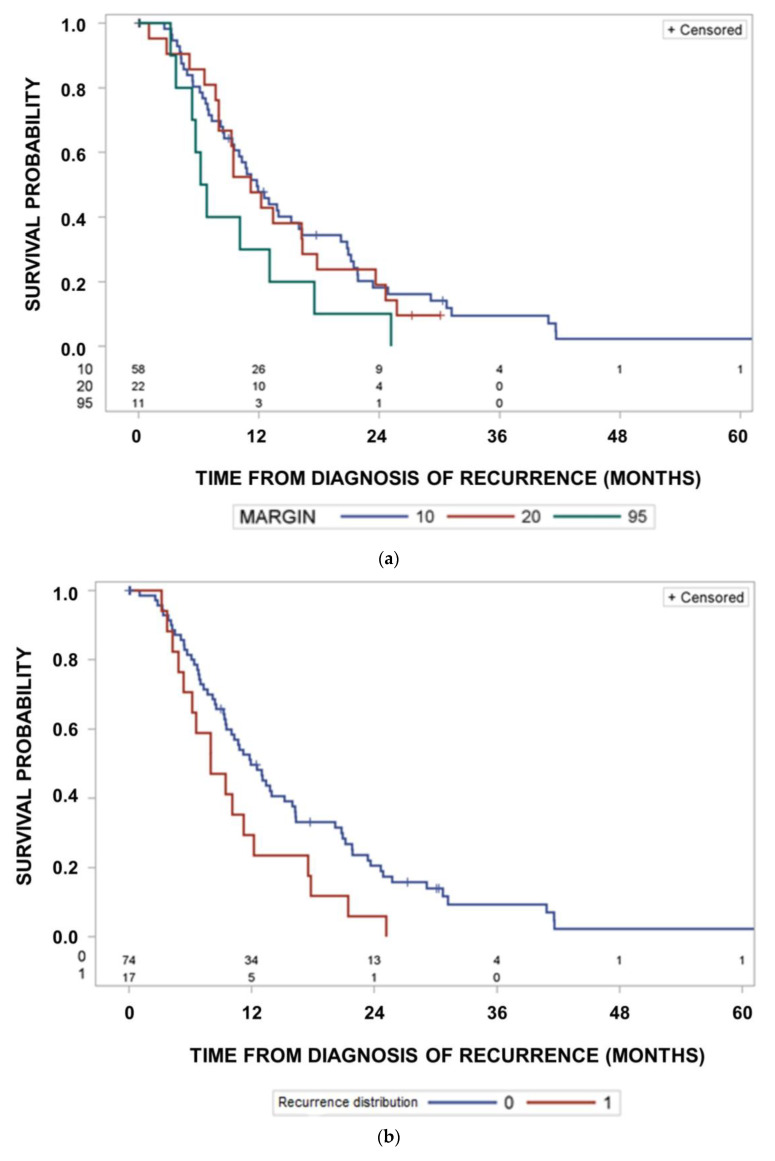
(**a**) Survival times from diagnosis of recurrence according to the margin at which the in-field criterion is fulfilled. Survival after recurrence differed significantly according to the location of recurrence (*p* = 0.1714, log-rank test). (**b**) Survival times from diagnosis of recurrence according to uni- or multifocality of the recurrence. Recurrence distribution: 0 unifocal, 1 multifocal. Survival curves differed significantly (*p* = 0.0294, log-rank test). (**c**) Survival times from diagnosis of recurrence according to the extent of salvage surgery. Resection status: 1, gross total resection; 2, partial resection or no resection. Survival curves differed significantly (*p* = 0.0069, log-rank test).

**Figure 6 cancers-15-02982-f006:**
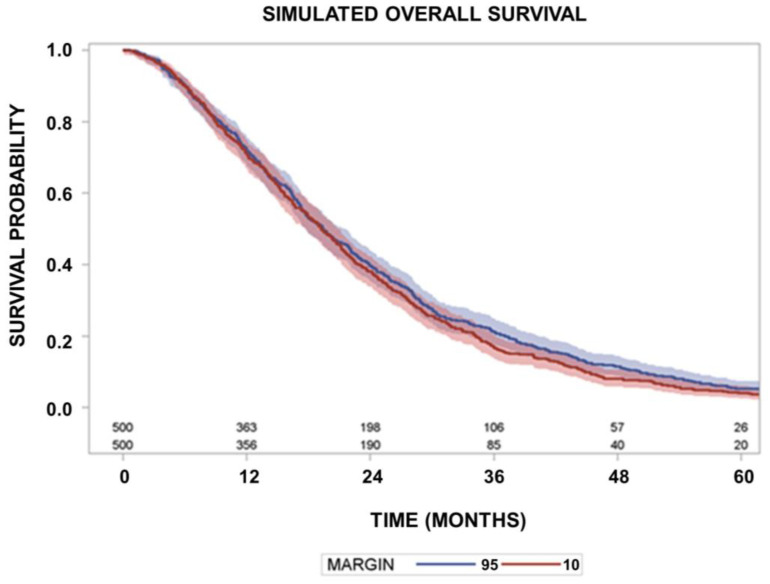
In silico simulated overall survival curves for patients irradiated with CTV margins of 10 mm or with conventional margins. Survival curves were simulated for dose distributions with conventional margins (blue) or a CTV margin of 10 mm (red) under the concurrent risks of recurrences at different locations, within 10 mm around the GTV, beyond 10 mm up to the 95% isodose of the conventional dose distribution, or out-field the 95% isodose. Survival curves did not differ even for large samples of 2 × 500 patients in this representative simulation results (*p* = 0.5333, log-rank test). Monte Carlo simulations were repeated 100 times assuming that small-volume radiotherapy with 10 mm CTV margins results in shorter times to recurrences located at distances >10 mm, with CTV margins smaller than the 95% isodose around the GTV by a median time of 5.5 months, and marginal recurrences located outside the 95% isodose by a median of 2.8 months in comparison to the delivered radiotherapy using conventional margins.

**Table 1 cancers-15-02982-t001:** Patient, tumor, and treatment characteristics analysed. Numbers given without units represent patient counts. Distribution of characteristics given with units are represented by median and range; the latter was given in brackets.

Characteristic		Patient Numbers
Age		59 y (37 y–80 y)
Sex	Male vs. female	50/41
Laterality of the tumor	Left vs. right, vs. bilateral	38/52/1
Location of the primary	frontal/temporal/parietal/parietooccipital/occipital/parietotemporal/other	30/22/13/7/7/6/6
Type of resection	gross total vs. partial resection	47/44
MGMT promoter methylation status	Unmethylated/methylated/not determined	57/30/4
Analysis of IDH-mutation status	Performed/not performed	71/20
Time from surgery to start of radiotherapy		28 d (7 d–97 d)
Volume of the postoperative GTV		26.5 cm^3^ (0.5 cm^3^–195.4 cm^3^)
Conventional fractionation vs. Hypofractionation of radiotherapy		85/6
Total dose	Conventional Hypofractionated	60 Gy (56 Gy–60 Gy) 40 Gy (40 Gy–40 Gy)
Concurrent single drug Temozolomide	yes/no	77/14
Concurrent Temozolomide and Lomustin	yes/no	5/86
TT-Fields consolidation	yes/no	13/78
Recurrence pattern	Uni- vs. multifocal recurrence	74/17
Re-resection as salvage therapy	No vs. complete vs. partial	58/14/19
Volume of the recurrence at diagnosis of recurrence		4.0 cm^3^ (0.1 cm^3^–115.8 cm^3^)

**Table 2 cancers-15-02982-t002:** Univariate analysis of characteristics related to in- or out-field location of the tumor progression. Univariate analysis of the association of factors from Table 1 with in- or out-field location of recurrences. *p*-values based on Fisher’s exact tests. Relative risks are given together with their 95% confidence intervals. Laterality: note that only two tumors were bilaterally located which were excluded in this row analysis. ^1^ odds ratio from logistic regression per increase in the natural logarithm of the respective volume by a value of 1. ^2^ odds ratio from logistic regression per day increase in the respective time interval. ^3^ chi-squared test.

		Relative Risk of Out-Field Recurrences	P (Exact Fisher Test)
Age	≤60 vs. >60	1.14 (0.43–3.03)	1.0000
Sex	Male vs. female	3.01 (0.90–10.06)	0.0790
Laterality of the tumor	left vs. right	1.37 (0.52–3.58)	0.5658
Location of the primary (over all locations)			0.2013
Resection status	Gross total vs. partial	1.69 (0.61–4.64)	0.3884
IDH-mutation status	Negative vs. not determined	1.69 (0.41–6.94)	0.7267
Temozolomide daily	Not concurrent vs. concurrent	1.50 (0.49–4.70)	0.4461
Lomustin and Temozolomide	Not concurrent vs. concurrent	0.76 (0.12–4.70)	0.5751
Radiation dose fractionation	Conventional vs. hypofractionated	0.92 (0.14–5.88)	1.0000
TT-Fields	No-consolidation vs. consolidation	2.17 (0.31 –15.19)	0.6829
MGMT methylation status	Methylated vs. unmethylated	1.06 (0.39–2.87)	1.0000
Recurrence pattern	Multi- vs. unifocal recurrence	4.35 (95%CI: 1.76–10.77)	0.0037
Re-resection as salvage therapy	No vs. incomplete	1.09 (0.26–4.48) 0.98 (0.30–3.26)	1.0000
Log (volume) of the recurrence		1.20 (0.85–1.73) ^1^	0.3210 ^3^
Log (volume) of the postoperative GTV (resection cavity + CM enhancement		0.86 (0.56–1.37) ^1^	0.5075 ^3^
Time from start of radiotherapy to recurrence		1.00 (0.998–1.002) ^2^	0.6904 ^3^
Waiting time between surgery and start of radiotherapy		1.00 (0.96–1.04) ^2^	0.8156 ^3^

## Data Availability

Data are unavailable due to privacy restrictions.

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
