# Peer review of "Location of Recurrences after Trimodality Treatment for Glioblastoma with Respect to the Delivered Radiation Dose Distribution and Its Influence on Prognosis"

_cancers, 2023, doi:10.3390/cancers15112982_

Round 1
Reviewer 1 Report
Dear Authors,
Your research field is crucial and front line for the battle of GBM. Your data can add additional knowledge to the major weapon of GBM, the irradiation. Nevertheless, the manuscript is written in a way that is not friendly to a reader who is not familiar with all those clinical definitions and there is no clear connection between your clinical observations and a possible translation for wet-lab researchers working for example in the field of translational cancer research.
Here are my comments and suggestions for the manuscript.
I found clear and useful the below:
Clear aim of the study at lines 86-91
Samples from a wide period of time (2007-2021)
Clear explanation of the confirmation of recurrence
Exclusion of IDH mutant gliomas
A clear conclusion: lines 382-385
Major suggestions
1. Line 21: Germany instead of German
2. Line 23: Rephrase the first sentence. Rephrase ‘This clinical examination’ to sound more formal and scientific.
3. Lines 24-25: Maybe make clearer that IMRT and VMAT are different kinds of radiation?
4. What the article proposes or suggests after all? Because at lines 28-29 is written:’Larger radiation therapy margins may decrease the proportion of out-28 field recurrences, but an effect on overall survival is highly questionable.’And at lines 415-417: ‘smaller CTV margins may allow for better protection of healthy tissue thus ensuring patients’ quality of life and leave therapeutic options for salvage RT in the yet unpreventable event of GBM recurrence’.
5. Line 48: Define the abbreviation GTV
6. Lines 51-56: delete them
7. Lines 57-58: delete the numbers
8. Line 80: Define the abbreviation PTV. In general make clear all our abbreviations because is difficult to follow, especially in the legends of the figures
9. Throughout the text the definitions ‘T1 weighted sequences’ and ‘T2/FLAIR’ are used. For someone not knowing those definitions is hard to follow. For example, at lines 137-138: ‘T2/FLAIR MRI high signal regions were included, if they were considered to represent regions of low grade tumor’. What information this T2/FLAIR MRI give us?
10. Define the definition of the isodose
11. Line 173 & 273: what means the number 1 at the ‘the 1-Kaplan-Meier probabilities’?
12. Why the figures are not in the proper order? Fig 1, 2, 3, 4, 5. If a figure has two parts a and b, then it should be one figure with two parts, not two different figures, like figure 4a and 4b in the text. It should be Figure 4 and then explain graph a and graph b.
13. Delete the word legend from the legends of most figures and tables.
14. Make sure for the proper appearance of the words ‘Figure 4a’ etc and that the title and the legend of the figure are aligned.
15. Title of figure 4a: what does the number means at ‘of 1-probability’?
16. In general, improve the clarity of the legends of all the figures
17. Aligned Table 1 with this title.
18. Maybe it will be helpful to broad the left grey column of the tables so that it is more friendly to the reader cause now it a little bit squished.
19. Give some space before line 233
20. Correct the word ‘radii’ at line 234
21. Maybe explain a little bit the definition multifocality. It is quite clear but when it is implied with other definitions it will be nice to be more clear.
22. The table at page 12 is a continuation of table 2 on page 11? Ideally it will be nice to be at the same page with clear legends.
23. Legend of figure 5b: ‘uni- or multi- focality’
24. Take care of the appearance of our text, like in lines 316-317, figure 4a and figure 6
25. Lines 368-369: explain a little bit more about the 5-ALA
26. Lines 415-417: As described above, it is a little bit confusing what is the message of the manuscript. Moreover, it will be nice if those results/conclusions could be given in a way that is clearer to molecular scientists working for example in translational cancer research. For example, if smaller margins are applied to protect the healthy tissue, what about the invading GBM cells, as GBM is highly disseminating? If recurrences occur within the gross tumor volume, this means that the left over cellular populations in the gross tumor region are important? For example, glioma stem cells could be one of those surviving populations? Are there any information of how multifocal recurrence occurs?
Minor suggestions
27. Space between ‘in11 patients’ at line 241
28. Legend of figure 2. Double space before ‘Margin 10mm,’
29. Line 271: ‘at margins of 10 mm. at a minimum margin > 10 mm’. There is a dot after mm
30. Line 353: Space between et al. and 2016
31. Line 415: Double space between for and better
Good luck.
Author Response
MS-N° |
cancers-2332904 |
Title: |
"Location of Recurrences after Trimodality Treatment For Glioblastoma with respect to the delivered radiation dose distribu-tion and its influence on prognosis"
|
Dear Referees,
we would like to thank you for the careful review of our manuscript “Location of Recurrences after Trimodality Treatment For Glio-blastoma with respect to the delivered radiation dose distribu-tion and its influence on prognosis”. We carried out a revision and hope we were able to meet all comments, suggestions and corrections. In the following we sequentially address all of the points raised in the, interactive review.
Reviewer 1
Dear Authors,
Your research field is crucial and front line for the battle of GBM. Your data can add additional knowledge to the major weapon of GBM, the irradiation. Nevertheless, the manuscript is written in a way that is not friendly to a reader who is not familiar with all those clinical definitions and there is no clear connection between your clinical observations and a possible translation for wet-lab researchers working for example in the field of translational cancer research.
Here are my comments and suggestions for the manuscript.
I found clear and useful the below:
Clear aim of the study at lines 86-91
Samples from a wide period of time (2007-2021)
Clear explanation of the confirmation of recurrence
Exclusion of IDH mutant gliomas
A clear conclusion: lines 382-385
“We kindly thank the reviewer for his opinion and valuable comments and at the same time tried hard to implement all suggestions and recommendations each addressed in the interactive review.”
Major suggestions
- Line 21: Germany instead of German
“Please excuse the lapsus calami which was corrected (page 1, line 21)”
- Line 23: Rephrase the first sentence. Rephrase ‘This clinical examination’ to sound more formal and scientific.
“The sentence was rephrased as recommended (page 1, line 23)”
- Lines 24-25: Maybe make clearer that IMRT and VMAT are different kinds of radiation?
“The paragraph was paraphrased to clarify the content (page 1, line 24-25)”
- What the article proposes or suggests after all? Because at lines 28-29 is written: ’Larger radiation therapy margins may decrease the proportion of out-28 field recurrences, but an effect on overall survival is highly questionable. ’And at lines 415-417: ‘smaller CTV margins may allow for better protection of healthy tissue thus ensuring patients’ quality of life and leave therapeutic options for salvage RT in the yet unpreventable event of GBM recurrence’.
“We thank the reviewer for this important comment (4.). We wanted to delineate that despite the possibly higher number of out-filed recurrences, the present results indicate that the classic GTV-CTV delineation strategy for postoperative RT of GBM should be reconsidered in favour of smaller CTV-margins. A reduction in CTV margins allows for better protection of healthy tissue thus ensuring patients’ quality of life and leave therapeutic options for salvage radiation therapy.”
- Line 48: Define the abbreviation GTV
“As recommended the term was defined (page 1, line 36)”
- Lines 51-56: delete them
“The respective sentences are deleted (page 2, line 51-56)”
- Lines 57-58: delete the numbers
“The respective numbers in the keyword section are deleted as advised (page 2, line 57-58)”
- Line 80: Define the abbreviation PTV. In general make clear all our abbreviations because is difficult to follow, especially in the legends of the figures
“As recommended the term was defined (page 2, former line 80, new line 76)”
- Throughout the text the definitions ‘T1 weighted sequences’ and ‘T2/FLAIR’ are used. For someone not knowing those definitions is hard to follow. For example, at lines 137-138: ‘T2/FLAIR MRI high signal regions were included, if they were considered to represent regions of low grade tumor’. What information this T2/FLAIR MRI give us?
“FLAIR-sequences are highly sensitive for white matter changes of the brain. In the setting of glioblastoma FLAIR-hyperintensities may not only be induced by edema, but these may also represent low grade tumor regions. Thus, FLAIR-sequences represent an important sequence in therapy planning.”
- Define the definition of the isodose
“As recommended the term isodose was defined (page 3, line 140)”
- Line 173 & 273: what means the number 1 at the ‘the 1-Kaplan-Meier probabilities’?
“Please excuse the lapsus calami. The number is deleted on the respective pages (page 4, line 170 and page 9, line 270)”
- Why the figures are not in the proper order? Fig 1, 2, 3, 4, 5. If a figure has two parts a and b, then it should be one figure with two parts, not two different figures, like figure 4a and 4b in the text. It should be Figure 4 and then explain graph a and graph b.
“Please excuse the mixed order of the figures. Now, all figures are reordered as chronologically discussed within the results section.”
- Delete the word legend from the legends of most figures and tables.
“As recommended the word is deleted from the legends.”
- Make sure for the proper appearance of the words ‘Figure 4a’ etc and that the title and the legend of the figure are aligned.
“The title and the legend of the figures are aligned as recommended.”
- Title of figure 4a: what does the number means at ‘of 1-probability’?
“Please excuse the lapsus calami. The number is deleted on the respective pages.”
- In general, improve the clarity of the legends of all the figures
“The legend of the figures is paraphrased to make it clearer as recommended.”
- Aligned Table 1 with this title.
“The title and the table are aligned as recommended.”
- Maybe it will be helpful to broad the left grey column of the tables so that it is more friendly to the reader cause now it a little bit squished.
“The columns of the tables are broadened for better readability.”
- Give some space before line 233
“For better readability, a line is inserted at the appropriate place.”
- Correct the word ‘radii’ at line 234
“The plural noun of the word radius is meant here.”
- Maybe explain a little bit the definition multifocality. It is quite clear but when it is implied with other definitions, it will be nice to be clearer.
“The word is explained as recommended (page 6, line 217-219).”
- The table at page 12 is a continuation of table 2 on page 11? Ideally it will be nice to be at the same page with clear legends.
“Correct. For a better readability, the title and legends of the table are set on the same page as recommended.”
- Legend of figure 5b: ‘uni- or multi- focality’
“The legend of the figure is paraphrased to make it clearer.”
- Take care of the appearance of our text, like in lines 316-317, figure 4a and figure 6
“Thank you for this comment. All shifted text parts are aligned.”
- Lines 368-369: explain a little bit more about the 5-ALA
“Gliolan (5-ALA) is approved in adults for the visualization of malignant tissue during surgery for malignant gliomas WHO grades III and IV. 5-ALA is a prodrug that is metabolized intracellularly to the fluorescent molecule PPIX. Tumor fluorescence is usually considerably higher than that of normal tissue (added to page 17, lines 387-391). Characteristically, the high contrast allows visualization of tumor tissue under blue-violet light. However, about half of tissue samples from the tumor boundary with no fluorescence signal contained infiltrative tumor tissue [reference 33]. It is known that the highest PPIX plasma level is reached four hours after oral administration of 20 mg/kg body weight 5-ALA HCl. Then PPIX plasma level decreases quickly over the next 20 hours and is undetectable 48 hours after administration.”
- Lines 415-417: As described above, it is a little bit confusing what is the message of the manuscript. Moreover, it will be nice if those results/conclusions could be given in a way that is clearer to molecular scientists working for example in translational cancer research. For example, if smaller margins are applied to protect the healthy tissue, what about the invading GBM cells, as GBM is highly disseminating? If recurrences occur within the gross tumor volume, this means that the left over cellular populations in the gross tumor region are important? For example, glioma stem cells could be one of those surviving populations? Are there any information of how multifocal recurrence occurs?
“This study shows that the concurrent risk of local recurrence within 1 cm margin to the gross tumor volume is on average dominant and occurs faster than recurrences at larger distances to the resection cavity or residual macroscopic tumor. As long as this risk remains as high as it is, therapies directed to cells more distant from the gross tumor will not be very successful. This also explains the failure of extensive resection attempts. As glioblastomas are heterogeneous, the hypothesis can be sustained from the present data that the most resistant population resides within the vicinity of the gross tumor.
Glioblastoma grows infiltrating in both, MRI-contrast-enhancing and non-enhancing regions. The inclusion of the non-contrast enhancing regions is based on stereotactic serial biopsy findings that showed that isolated tumor cells can be found in the tumor surrounding edema regions as far as hyperintensities on T2 weighted MRI exist [reference 5]. Former authors demonstrated that histopathologic features of glioblastoma are differentially expressed between MRI-contrast-enhancing and non-enhancing tumor components [Barajas RF, Phillips JJ, Parvataneni R et al (2012) Regional variation in histopathologic features of tumor specimens from treatment-naive glioblastoma correlates with anatomic and physiologic MR Imaging. Neuro Oncol 14:942–954]. According to these findings in the non-enhancing tumor components, which contain microscopic infiltrating tumor and vasogenic edema, lack some of the characteristic histopathologic features of glioblastoma. Therefore, these tumor regions may respond differently to radiation when compared to MRI-contrast-enhancing tissues [Barajas RF et al, 2012]. It may be assumed that for these isolated cells a lower dose for tumor control is necessary.
Different tumor cell foci can be genetically heterogeneous, can share only half of their mutations and can acquire additional mutations by parallel genetic evolution [Abou-El-Ardat K, Seifert M, Becker K et al. Comprehensive molecular characterization of multifocal glioblastoma proves its monoclonal origin and reveals novel insights into clonal evolution and heterogeneity of glioblastomas. Neuro Oncol. 2017;19:546-557]. Kiesel et al analysed in their prospective study tissue samples from 5-ALA fluorescence guided glioblastoma resection with differing 5-ALA staining. Tumor cells in samples with no fluorescence showed less mitotic activity and less cell density than in 5-ALA positive zones [Kiesel B, Mischkulnig M, Woehrer A et al. Systematic histopathological analysis of different 5-aminolevulinic acid-induced fluorescence levels in newly diagnosed glioblastomas. J Neurosurg. 2018 Aug;129(2):341-353]. In addition there is evidence, that 5-ALA labelling corresponds with 11C Methionine PET uptake [Shimizu K, Tamura K, Hara S, et al. Correlation of Intraoperative 5-ALA-Induced Fluorescence Intensity and Preoperative 11C-Methionine PET Uptake in Glioma Surgery. Cancers (Basel). 2022;14:1449]. High volume of extravascular extracellular space by T1 dynamic contrast-enhanced MRI corresponds to increased mitotic activity of glioblastoma cells [Mills SJ, du Plessis D, Pal P et al. Mitotic Activity in Glioblastoma Correlates with Estimated Extravascular Extracellular Space Derived from Dynamic Contrast-Enhanced MR Imaging. AJNR Am J Neuroradiol. 2016;37:811-7]. In conclusion, these data support the hypotheses, that the most resistant parts of the tumor are clinically detectable and are included into the target volume within 1 cm margin.
In the first approaches of radiation therapy of glioblastoma, radiation was delivered as whole-brain irradiation. More and more radiation was conducted as partial brain irradiation in which only the highest risk areas are treated. At the same time there persisted transatlantic differences of standard margin recipes between Europe and USA [Reference 1 and 2]. However, according to both consensus guidelines, a slightly lower dose, such as 54–55.8 Gy can be applied when the tumor volume is very large, for neurocognitive structures such as the hippocampi or brain stem and for grade 3 astrocytoma for sufficient tumor control. At the same time, in the event of recurrence, there are more therapy options such as a salvage re-irradiation, when smaller margins are applied in the first series (added to the discussion section on page 16-18).”
Minor suggestions
- Space between ‘in11 patients’ at line 241
“Thank you for this comment, space is inserted at the appropriate place (page 6, line 239).”
- Legend of figure 2. Double space before ‘Margin 10mm,’
“Thank you for this comment, the double space is deleted.”
- Line 271: ‘at margins of 10 mm. at a minimum margin > 10 mm’. There is a dot after mm
“Thank you for this comment, the dot is deleted.”
- Line 353: Space between et al. and 2016
“Thank you for this comment, space is inserted at the appropriate place (discussion section on page 15, line 365.”
- Line 415: Double space between for and better
“Thank you for this comment, the double space is deleted.”
Good luck.
“We thank you for the thorough review and valuable comments that significantly improved the manuscript.”
Reviewer 2
In the present study, Guberina et al present a retrospective analysis complemented by Monte Carlo simulation concerning the margins of radiotherapy after Glioblastoma resection. The study demonstrates that the margins of radiotherapy might be reduced to 10 mm without causing significant decrease of recurrence free survival or overall survival. This is important, as it leaves more options for the treatment at the time of tumor recurrence
Please find my comments below:
The manuscript is nicely written, only minor corrections are necessary, although i think it could be shortened a bit.
“We thank you for the review and tried to explain some terms to improve readability.”
The main shortcoming is the lack of subgroup analysis (initial GTR vs. STR in particular), this was stated by the authors.
The authors may reconsider providing specific codes of particular statistical software and describe the methods they used instead.
“We thank the reviewer for this point. Upon request all specific codes of SAS may be provided to interested readers.”
The figures need some editing (e.g. Legend Figure 5b: Uni_Multifocal)
“The legend of the figures are edited where necessary and paraphrased to make it clearer.”

Reviewer 2 Report
In the present study, Guberina et al present a retrospective analysis complemented by Monte Carlo simulation concerning the margins of radiotherapy after Glioblastoma resection. The study demonstrates that the margins of radiotherapy might be reduced to 10 mm without causing significant decrease of recurrence free survival or overall survival. This is important, as it leaves more options for the treatment at the time of tumor recurrence
Please find my comments below:
The manuscript is nicely written, only minor corrections are necessary, although i think it could be shortened a bit.
The main shortcoming is the lack of subgroup analysis (initial GTR vs. STR in particular), this was stated by the authors.
The authors may reconsider providing specific codes of particular statistical software and describe the methods they used instead .
The figures need some editing (e.g. Legend Figure 5b: Uni_Multifocal)
Author Response
MS-N° |
cancers-2332904 |
Title: |
"Location of Recurrences after Trimodality Treatment For Glioblastoma with respect to the delivered radiation dose distribution and its influence on prognosis"
|
Dear Referees,
we would like to thank you for the careful review of our manuscript “Location of Recurrences after Trimodality Treatment For Glio-blastoma with respect to the delivered radiation dose distribu-tion and its influence on prognosis”. We carried out a revision and hope we were able to meet all comments, suggestions and corrections. In the following we sequentially address all of the points raised in the, interactive review.
Reviewer 1
Dear Authors,
Your research field is crucial and front line for the battle of GBM. Your data can add additional knowledge to the major weapon of GBM, the irradiation. Nevertheless, the manuscript is written in a way that is not friendly to a reader who is not familiar with all those clinical definitions and there is no clear connection between your clinical observations and a possible translation for wet-lab researchers working for example in the field of translational cancer research.
Here are my comments and suggestions for the manuscript.
I found clear and useful the below:
Clear aim of the study at lines 86-91
Samples from a wide period of time (2007-2021)
Clear explanation of the confirmation of recurrence
Exclusion of IDH mutant gliomas
A clear conclusion: lines 382-385
“We kindly thank the reviewer for his opinion and valuable comments and at the same time tried hard to implement all suggestions and recommendations each addressed in the interactive review.”
Major suggestions
- Line 21: Germany instead of German
“Please excuse the lapsus calami which was corrected (page 1, line 21)”
- Line 23: Rephrase the first sentence. Rephrase ‘This clinical examination’ to sound more formal and scientific.
“The sentence was rephrased as recommended (page 1, line 23)”
- Lines 24-25: Maybe make clearer that IMRT and VMAT are different kinds of radiation?
“The paragraph was paraphrased to clarify the content (page 1, line 24-25)”
- What the article proposes or suggests after all? Because at lines 28-29 is written: ’Larger radiation therapy margins may decrease the proportion of out-28 field recurrences, but an effect on overall survival is highly questionable. ’And at lines 415-417: ‘smaller CTV margins may allow for better protection of healthy tissue thus ensuring patients’ quality of life and leave therapeutic options for salvage RT in the yet unpreventable event of GBM recurrence’.
“We thank the reviewer for this important comment (4.). We wanted to delineate that despite the possibly higher number of out-filed recurrences, the present results indicate that the classic GTV-CTV delineation strategy for postoperative RT of GBM should be reconsidered in favour of smaller CTV-margins. A reduction in CTV margins allows for better protection of healthy tissue thus ensuring patients’ quality of life and leave therapeutic options for salvage radiation therapy.”
- Line 48: Define the abbreviation GTV
“As recommended the term was defined (page 1, line 36)”
- Lines 51-56: delete them
“The respective sentences are deleted (page 2, line 51-56)”
- Lines 57-58: delete the numbers
“The respective numbers in the keyword section are deleted as advised (page 2, line 57-58)”
- Line 80: Define the abbreviation PTV. In general make clear all our abbreviations because is difficult to follow, especially in the legends of the figures
“As recommended the term was defined (page 2, former line 80, new line 76)”
- Throughout the text the definitions ‘T1 weighted sequences’ and ‘T2/FLAIR’ are used. For someone not knowing those definitions is hard to follow. For example, at lines 137-138: ‘T2/FLAIR MRI high signal regions were included, if they were considered to represent regions of low grade tumor’. What information this T2/FLAIR MRI give us?
“FLAIR-sequences are highly sensitive for white matter changes of the brain. In the setting of glioblastoma FLAIR-hyperintensities may not only be induced by edema, but these may also represent low grade tumor regions. Thus, FLAIR-sequences represent an important sequence in therapy planning.”
- Define the definition of the isodose
“As recommended the term isodose was defined (page 3, line 140)”
- Line 173 & 273: what means the number 1 at the ‘the 1-Kaplan-Meier probabilities’?
“Please excuse the lapsus calami. The number is deleted on the respective pages (page 4, line 170 and page 9, line 270)”
- Why the figures are not in the proper order? Fig 1, 2, 3, 4, 5. If a figure has two parts a and b, then it should be one figure with two parts, not two different figures, like figure 4a and 4b in the text. It should be Figure 4 and then explain graph a and graph b.
“Please excuse the mixed order of the figures. Now, all figures are reordered as chronologically discussed within the results section.”
- Delete the word legend from the legends of most figures and tables.
“As recommended the word is deleted from the legends.”
- Make sure for the proper appearance of the words ‘Figure 4a’ etc and that the title and the legend of the figure are aligned.
“The title and the legend of the figures are aligned as recommended.”
- Title of figure 4a: what does the number means at ‘of 1-probability’?
“Please excuse the lapsus calami. The number is deleted on the respective pages.”
- In general, improve the clarity of the legends of all the figures
“The legend of the figures is paraphrased to make it clearer as recommended.”
- Aligned Table 1 with this title.
“The title and the table are aligned as recommended.”
- Maybe it will be helpful to broad the left grey column of the tables so that it is more friendly to the reader cause now it a little bit squished.
“The columns of the tables are broadened for better readability.”
- Give some space before line 233
“For better readability, a line is inserted at the appropriate place.”
- Correct the word ‘radii’ at line 234
“The plural noun of the word radius is meant here.”
- Maybe explain a little bit the definition multifocality. It is quite clear but when it is implied with other definitions, it will be nice to be clearer.
“The word is explained as recommended (page 6, line 217-219).”
- The table at page 12 is a continuation of table 2 on page 11? Ideally it will be nice to be at the same page with clear legends.
“Correct. For a better readability, the title and legends of the table are set on the same page as recommended.”
- Legend of figure 5b: ‘uni- or multi- focality’
“The legend of the figure is paraphrased to make it clearer.”
- Take care of the appearance of our text, like in lines 316-317, figure 4a and figure 6
“Thank you for this comment. All shifted text parts are aligned.”
- Lines 368-369: explain a little bit more about the 5-ALA
“Gliolan (5-ALA) is approved in adults for the visualization of malignant tissue during surgery for malignant gliomas WHO grades III and IV. 5-ALA is a prodrug that is metabolized intracellularly to the fluorescent molecule PPIX. Tumor fluorescence is usually considerably higher than that of normal tissue (added to page 17, lines 387-391). Characteristically, the high contrast allows visualization of tumor tissue under blue-violet light. However, about half of tissue samples from the tumor boundary with no fluorescence signal contained infiltrative tumor tissue [reference 33]. It is known that the highest PPIX plasma level is reached four hours after oral administration of 20 mg/kg body weight 5-ALA HCl. Then PPIX plasma level decreases quickly over the next 20 hours and is undetectable 48 hours after administration.”
- Lines 415-417: As described above, it is a little bit confusing what is the message of the manuscript. Moreover, it will be nice if those results/conclusions could be given in a way that is clearer to molecular scientists working for example in translational cancer research. For example, if smaller margins are applied to protect the healthy tissue, what about the invading GBM cells, as GBM is highly disseminating? If recurrences occur within the gross tumor volume, this means that the left over cellular populations in the gross tumor region are important? For example, glioma stem cells could be one of those surviving populations? Are there any information of how multifocal recurrence occurs?
“This study shows that the concurrent risk of local recurrence within 1 cm margin to the gross tumor volume is on average dominant and occurs faster than recurrences at larger distances to the resection cavity or residual macroscopic tumor. As long as this risk remains as high as it is, therapies directed to cells more distant from the gross tumor will not be very successful. This also explains the failure of extensive resection attempts. As glioblastomas are heterogeneous, the hypothesis can be sustained from the present data that the most resistant population resides within the vicinity of the gross tumor.
Glioblastoma grows infiltrating in both, MRI-contrast-enhancing and non-enhancing regions. The inclusion of the non-contrast enhancing regions is based on stereotactic serial biopsy findings that showed that isolated tumor cells can be found in the tumor surrounding edema regions as far as hyperintensities on T2 weighted MRI exist [reference 5]. Former authors demonstrated that histopathologic features of glioblastoma are differentially expressed between MRI-contrast-enhancing and non-enhancing tumor components [Barajas RF, Phillips JJ, Parvataneni R et al (2012) Regional variation in histopathologic features of tumor specimens from treatment-naive glioblastoma correlates with anatomic and physiologic MR Imaging. Neuro Oncol 14:942–954]. According to these findings in the non-enhancing tumor components, which contain microscopic infiltrating tumor and vasogenic edema, lack some of the characteristic histopathologic features of glioblastoma. Therefore, these tumor regions may respond differently to radiation when compared to MRI-contrast-enhancing tissues [Barajas RF et al, 2012]. It may be assumed that for these isolated cells a lower dose for tumor control is necessary.
Different tumor cell foci can be genetically heterogeneous, can share only half of their mutations and can acquire additional mutations by parallel genetic evolution [Abou-El-Ardat K, Seifert M, Becker K et al. Comprehensive molecular characterization of multifocal glioblastoma proves its monoclonal origin and reveals novel insights into clonal evolution and heterogeneity of glioblastomas. Neuro Oncol. 2017;19:546-557]. Kiesel et al analysed in their prospective study tissue samples from 5-ALA fluorescence guided glioblastoma resection with differing 5-ALA staining. Tumor cells in samples with no fluorescence showed less mitotic activity and less cell density than in 5-ALA positive zones [Kiesel B, Mischkulnig M, Woehrer A et al. Systematic histopathological analysis of different 5-aminolevulinic acid-induced fluorescence levels in newly diagnosed glioblastomas. J Neurosurg. 2018 Aug;129(2):341-353]. In addition there is evidence, that 5-ALA labelling corresponds with 11C Methionine PET uptake [Shimizu K, Tamura K, Hara S, et al. Correlation of Intraoperative 5-ALA-Induced Fluorescence Intensity and Preoperative 11C-Methionine PET Uptake in Glioma Surgery. Cancers (Basel). 2022;14:1449]. High volume of extravascular extracellular space by T1 dynamic contrast-enhanced MRI corresponds to increased mitotic activity of glioblastoma cells [Mills SJ, du Plessis D, Pal P et al. Mitotic Activity in Glioblastoma Correlates with Estimated Extravascular Extracellular Space Derived from Dynamic Contrast-Enhanced MR Imaging. AJNR Am J Neuroradiol. 2016;37:811-7]. In conclusion, these data support the hypotheses, that the most resistant parts of the tumor are clinically detectable and are included into the target volume within 1 cm margin.
In the first approaches of radiation therapy of glioblastoma, radiation was delivered as whole-brain irradiation. More and more radiation was conducted as partial brain irradiation in which only the highest risk areas are treated. At the same time there persisted transatlantic differences of standard margin recipes between Europe and USA [Reference 1 and 2]. However, according to both consensus guidelines, a slightly lower dose, such as 54–55.8 Gy can be applied when the tumor volume is very large, for neurocognitive structures such as the hippocampi or brain stem and for grade 3 astrocytoma for sufficient tumor control. At the same time, in the event of recurrence, there are more therapy options such as a salvage re-irradiation, when smaller margins are applied in the first series (added to the discussion section on page 16-18).”
Minor suggestions
- Space between ‘in11 patients’ at line 241
“Thank you for this comment, space is inserted at the appropriate place (page 6, line 239).”
- Legend of figure 2. Double space before ‘Margin 10mm,’
“Thank you for this comment, the double space is deleted.”
- Line 271: ‘at margins of 10 mm. at a minimum margin > 10 mm’. There is a dot after mm
“Thank you for this comment, the dot is deleted.”
- Line 353: Space between et al. and 2016
“Thank you for this comment, space is inserted at the appropriate place (discussion section on page 15, line 365.”
- Line 415: Double space between for and better
“Thank you for this comment, the double space is deleted.”
Good luck.
“We thank you for the thorough review and valuable comments that significantly improved the manuscript.”
Reviewer 2
In the present study, Guberina et al present a retrospective analysis complemented by Monte Carlo simulation concerning the margins of radiotherapy after Glioblastoma resection. The study demonstrates that the margins of radiotherapy might be reduced to 10 mm without causing significant decrease of recurrence free survival or overall survival. This is important, as it leaves more options for the treatment at the time of tumor recurrence
Please find my comments below:
The manuscript is nicely written, only minor corrections are necessary, although i think it could be shortened a bit.
“We thank you for the review and tried to explain some terms to improve readability.”
The main shortcoming is the lack of subgroup analysis (initial GTR vs. STR in particular), this was stated by the authors.
The authors may reconsider providing specific codes of particular statistical software and describe the methods they used instead.
“We thank the reviewer for this point. Upon request all specific codes of SAS may be provided to interested readers.”
The figures need some editing (e.g. Legend Figure 5b: Uni_Multifocal)
“The legend of the figures are edited where necessary and paraphrased to make it clearer.”
